# Explainable Attribution using Additive Gaussian Processes

**Xiaoyu Lu**                                                                    LUXIAOYU@AMAZON.COM
*Amazon*

**Alexis Boukouvalas**                                              ALEXISBOUK@DEEPMIND.COM
*Google Deepmind*

**James Hensman**                                              JAMESHENSMAN@MICROSOFT.COM
*Microsoft Research*

## Abstract

With the advances of computational power, there has been a rapid development in complex systems to predict certain outputs for industrial problems. Attributing outputs to input features, or output changes to input or system changes has been a critical and challenging problem in many real world applications. In industrial settings, a system could be a chain of large scale models or simulators, or a combination of both that are black-box and hard to interpret. The inputs to a system can change over time due to more information becoming available, the system itself can also be updated as new models or simulators get deployed or retrained. Understanding how system behaviours change provides invaluable insights in the black-box systems and aids for critical downstream business decision making. Attribution is the framework that tackles these problems. However, balancing explainability versus accuracy remains a challenging problem. On one hand, attribution methods based on black-box models are flexible enough to represent the systems but lack explainability; simpler models such as linear regression are interpretable enough, however, they lack the ability to represent the systems well. In this paper, we propose an explainable attribution framework based on additive Gaussian processes that can be applied to three major types of attribution tasks commonly seen in practice, where we demonstrate with a toy example for each use case.

## 1. Introduction

Many science and engineering applications involve black-box systems that predict outputs given multiple inputs. Such systems can be a complex combination of sub-systems that consist of multiple large-scale machine learning models, simulators or engineering systems. Given multiple sources of input features, the systems forecast predictions for the output metrics. The predictions of the outputs change from time to time due to input changes as new training data becomes available, or due to system updates which may involve new model deployment, model re-training or engineering updates. Attributing the outputs to inputs provides valuable insights in understanding how the complicated systems depend on various sources of input features. Attributing the evolution of output changes provides critical value in understanding the system behaviour, sensitivity of the system outputs to input changes, and the impact of system updates, which are the foundation for the follow-up decision making processes. Having an explainable attribution framework that is capable to address these business challenges provides multiple benefits, including feature selection based on attribution; anomaly detection; system trouble shooting when the learned input-output

relationship is against experts' domain knowledge; guiding system updates, such as model reversion or deprecation when they contribute negatively to the desired system behaviour.

There exists a range of attribution methods among which a coalition game theoretical methodology, namely Shapley values (Shapley et al., 1953), has become a popular method for feature attribution in machine learning, due to its advantageous properties such as efficiency, additivity, symmetry, dummy, etc. (Lundberg and Lee, 2017; Štrumbelj and Kononenko, 2014; Lundberg et al., 2019; Chau et al., 2022, 2023). The main idea behind it is to build an underlying model to mimic the the systems (i.e., input-output relationships) and compute an average of multiple counterfactual effects through keeping certain features in while leaving out other features. While Shapley value is analytic when using linear models, for most black-box models such as tree-based or deep learning models, Shapley values do not have a closed-form solution. In such cases, it typically requires Monte Carlo sampling estimation (Štrumbelj and Kononenko, 2014) where the computation time increases exponentially with the number of features, limiting its usability for large or high-dimensional systems. On the other hand, unlike supervised learning tasks, attribution usually does not have a ground truth in real-world applications and verifying the attribution results is difficult. Providing reliable and trustworthy attribution is a challenging problem, which motivates us to build explainable attribution with insights into the internal mechanisms of the systems.

An explainable model that explains the system behaviour is one of the key factors in attribution. Recently, Lu et al. (2022) proposed an explainable model based on Gaussian processes with Orthogonal Additive Kernel (OAK) that provides low-dimensional parsimonious representations while retaining competitive predictive performance. Given its explainability, uncertainty quantification capability and efficient Shapley value calculation, we propose to use OAK as the underlying model for 3 commonly seen attribution tasks: (1) attributing output to inputs; (2) attributing output to inputs with system change; (3) attributing output changes to input changes. Our contributions include: for (1) we propose to use Sobol' indices (Owen, 2014), and we provide closed-form expressions for the Sobol' indices and Shapley value under the OAK Gaussian process model. For (2) We propose to model the dataset shift using an additional indicator input variable. We provide a kernel under the OAK framework for handling this additional (binary) input variable in a way that is consistent with continuous OAK kernels. For (3) we provide analytic expressions for Shapley values under the OAK model to explains changes in the output. We also demonstrate how we can leverage uncertainty quantification from first principle to provide confidence intervals of attribution, which are essential in risk-aware decision making processes.

The paper is organised as follows: in Section 2 we briefly recap on the OAK model. We link OAK with Shapley value attribution in Section 3. In Section 4 we discuss different types of attribution tasks commonly seen in practice and demonstrate how we can address these problems leveraging the OAK model. Finally we conclude in Section 5.

## 2. Orthogonal Additive Gaussian Processes Recap

We give a brief recap on the Gaussian processes with Orthogonal Additive Kernel (OAK) (Lu et al., 2022). Suppose we have output $y$ as a function of $d$-dimensional input features $\mathbf{x} := (x_1, \cdots x_d)$, Duvenaud et al. (2011) considers building a GP model with the additive

structure:

$$f(\mathbf{x}) = f_1(x_1) + f_2(x_2) + \cdots + f_{12}(x_1, x_2) + \cdots + f_{12\ldots d}(x_1, x_2, \cdots x_d). \tag{1}$$

Suppose the *base kernel* for each dimension $i \in \{1...d\}$ is $k_i(x_i, x_i')$, Lu et al. (2022) constrained each functional component $f_i$ with a modified constrained kernel $\tilde{k}_i(x_i, x_i')$, such that: $\int_{\mathcal{X}_i} f_i(x_i) p_i(x_i) dx_i = 0$ for $i \in [d]$, where $[d]$ denotes all possible subsets of an index set $\{1, \ldots, d\}$, $\mathcal{X}_i$ and $p_i$ are the sample space and the density for input feature $x_i$. The $n^{th}$ order additive kernel in Duvenaud et al. (2011) are then replaced with the constrained kernel:

$$k_{add_n}(x, x') = \sigma_n^2 \sum_{1 \leq i_1 \leq i_2 \leq \cdots \leq i_d \leq d} \left[ \prod_{l=1}^{n} \tilde{k}_{i_l}(x_{i_l}, x_{i_l}') \right]. \tag{2}$$

With this construction, Lu et al. (2022) demonstrated its competitive model predictive performance, parsimonious and low-dimensional representation on a range of regression and classification problems.

## 3. Sobol' Index and Shapley Value

Sobol' index, defined as $S_u := \mathbb{V}_x[f_u(\mathbf{x})]$ for feature set $u$, is a measure that quantifies the contribution of each feature set $u$ to the variance of the overall function. The Sobol' index of the OAK model is approximated by the best linear predictor which is the posterior mean of the GP $m_u(x)$:

$$\frac{var_x[f_u(x)]}{var_x[f(x)]} \approx \frac{var_x[m_u(x)]}{var_x[m(x)]} \tag{3}$$

where the variance in $f$ is ignored. It has been proven that the Sobol' index on the GP posterior mean is analytic for the OAK model thanks to the construction of the constrained kernels (see Section 4 of Lu et al. (2022)). Shapley value, as one of the most popular metrics for attribution, is defined as

$$\phi_j = \frac{1}{d} \sum_{u \subseteq [d] \setminus \{j\}} \binom{d-1}{|u|}^{-1} (\nu(u \cup \{j\}) - \nu(u)) \tag{4}$$

for feature $j$, where $\nu$ is some value function. Owen (2014) showed that the Sobol' index $S_u$ derived from an ANOVA decomposition satisfies $\phi_j = \sum_{u \subseteq [d], j \in u} \frac{S_u}{|u|}$ where $|u|$ is the cardinality of set $u$. Intuitively, the Shapley value for an input $j$ includes the variance components of all factors where input $j$ appears, with the weight inversely proportional to the number of factors in a variance component. Since the functions in the OAK model are precisely the components of the functional ANOVA decomposition, it follows that the OAK model has an analytic form of the Shapley value where no Monte Carlo sampling is required.

## 4. Attribution

We describe a few commonly seen attribution tasks and illustrate with synthetic examples how we can leverage the OAK model for attribution.

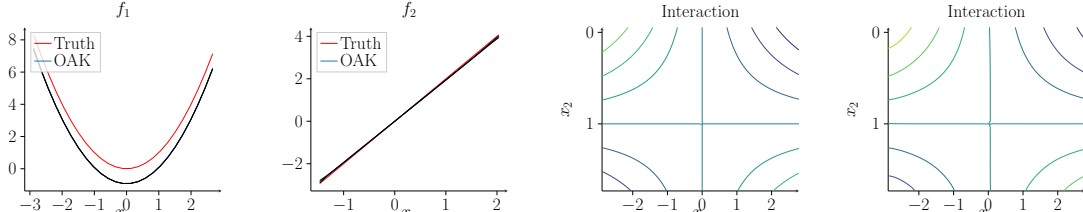

Figure 1: True and learned functional decomposition. (a) $f_1$; (b) $f_2$; (c) true interaction between $x_1$ and $x_2$; (d): learned interaction between $x_1$ and $x_2$. Note that the constant gap between the truth and OAK in the first plot is expected and is captured with the constant kernel.

## 4.1. Attributing Output to Inputs

This is one of the commonly seen attribution tasks where the goal is to explain the output predictions to the input features. This can be used for e.g., gaining insights into the systems, optimising for feature selections and quantifying the importance of features. We illustrate with a two-dimensional problem where the inputs $x_1$, $x_2$ and output $y$ are generated with

$$x_1 \sim \mathcal{N}(0,1), \quad x_2 \sim \mathcal{N}(0,0.25), \quad y = x_1^2 + 2x_2 - x_1x_2 + \epsilon \quad (5)$$

where $\epsilon \sim \mathcal{N}(0,0.01)$. Fitting the OAK model with 1000 training data yields the learned functional decomposition shown in Figure 1. We can observe that OAK is able to recover the correct decomposition, which is a pre-requisite for accurate attribution tasks. The Shapley value can be computed exactly as shown in Figure 2 (left). We observe that $x_1$ has a bigger impact on the output predictions, this is as expected since the quadratic form of $f_1$ has stronger impact than the linear form of $f_2$.

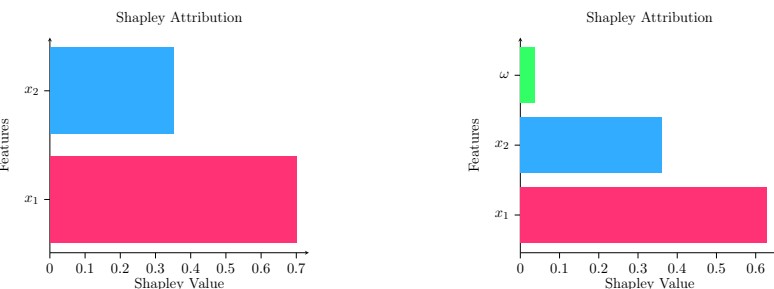

Figure 2: Shapley value attribution for the synthetic example, left: attributing output to inputs; right: attributing output to inputs with mechanism change.

## 4.2. Attributing Output to Inputs with System Change

In practice, often not only the inputs affect the output, system updates can also have an impact. System updates can be due to new model launches in certain components of the systems, model re-training, and other engineering infrastructure updates that affect the system output predictions. In this case, the underlying functional mechanism $f$ that generates the two set of outputs are different. To gain insights into how the system update affect the

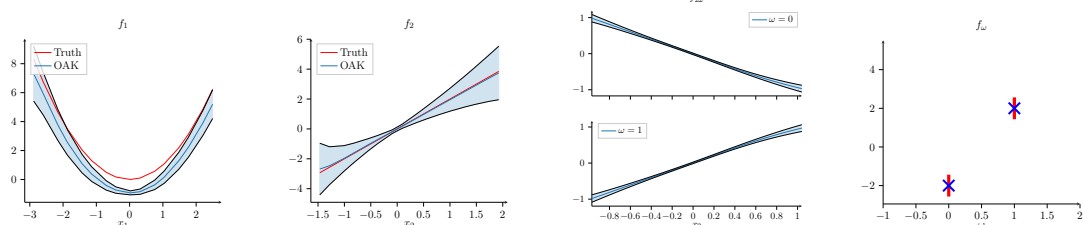

Figure 3: True and learned functional decomposition with mechanism change. (a) $f_1$; (b) $f_2$; (c) interaction between $x_2$ and $w$; (d): $f_w$.

output, we introduce an additional binary feature $\omega$ to indicate whether the data is coming from the old system ($\omega = 0$) or the new system ($\omega = 1$). Take the toy example above, for any input feature $(x_1, x_2, \omega)$, we decompose the function as

$$f(x_1, x_2, \omega) = f_1(x_1) + f_2(x_2) + f_\omega(\omega) + f_{12}(x_1, x_2) + f_{1\omega}(x_1, \omega) + f_{2\omega}(x_2, \omega) + f_{12\omega}(x_1, x_2, \omega).$$

Similarly as the input features $x_1, x_2$, we propose an orthogonal binary kernel to model the indicator variable $\omega$, whose Sobol' index and Shapley value are also analytic (see details in Appendix A). Importantly, one can examine and visualise $f_w$ and its interactions with the input features $f_{1\omega}(x_1, \omega), f_{2\omega}(x_2, \omega), f_{12\omega}(x_1, x_2, \omega)$ to understand how the mechanism has affected the output generating process through the inputs, offering deeper and insightful explanation. For illustration, we modify the toy example with two different data generating process through adding additional shift and interaction vis $\omega$:

$$y = x_1^2 + 2x_2 - x_1x_2 + 4(\omega - 0.5) + 2x_2(w - 0.5) + \epsilon, \qquad \omega \in \{0, 1\} \tag{6}$$

where $\omega$ changes the slope of $f_2$ through interacting with $x_2$ and shifts the output up or down. We generate half of the data with $\omega = 0$ and the other half with $\omega = 1$. The learned functional decomposition can be found in Figure 3, where we observe that OAK correctly recovered the ground truth data generating process: similar $f_1$ and $f_2$ are learned compared with Figure 1, except with bigger variance; the interaction between $x_2$ and $\omega$ is correctly learned where $\omega = 1$ reduces the slope of $x_2$ by 1 whereas $\omega = 0$ increases it; the main effect of $\omega$ is also correctly learned with a treatment effect of $2 - (-2) = 4$. Shapley value attribution can be found in Figure 2 (right) where we observe $x_1$ remains the main contributor.

### 4.3. Attributing Output Changes to Input Changes

Another typical attribution task is to attribute the evolution of output changes due to input data updates. This is common when the systems take more recent input data to make updated output predictions over time. The goal is to attribute the delta of the output to each of the input features. Take a two-dimensional toy example with the following decomposition:

$$f(x_1, x_2) = f_1(x_1) + f_2(x_2) + f_{12}(x_1, x_2) + \mathbb{E}_{x_1, x_2}[f(x_1, x_2)], \tag{7}$$

let $\mathbf{x}^1 := (x_1^1, x_2^1)$ and $\mathbf{x}^2 := (x_1^2, x_2^2)$ be features for a single data point for datasets 1 and 2 respectively, where the superscripts represent different datasets and the subscripts represent different features. Suppose the output $f(x_1^1, x_2^1)$ has changed to $f(x_1^2, x_2^2)$, we aim to attribute

this change to each of its input feature, such that Shapley value efficiency is satisfied, i.e., $\psi_1(\mathbf{x}^1, \mathbf{x}^2) + \psi_2(\mathbf{x}^1, \mathbf{x}^2) = f(x_1^2, x_2^2) - f(x_1^1, x_2^1)$, where $\psi$ represent the Shapley values.

The Shapley value defined in Section 4.1 and 4.2 are based on the variance of each functional component. For this particular attribution task, it is often desirable to have attribution on the data point level granularity which can be used to deep dive system evolution (i.e., explaining attribution of the delta in the output for each data point). Instead of using the Sobol' index as the value function as in Section 3, we propose to use

$$\nu_x(u) = \int f(x_1, x_2, \cdots, x_d) d\mathbb{P}_{-u} \tag{8}$$

where $f$ takes the form of the OAK model. The value functions for each of the subset of $\{1, 2\}$ are therefore

$$\nu_x(\{1\}) = f_1(x_1), \quad \nu_x(\{2\}) = f_2(x_2), \quad \nu_x(\{1, 2\}) = f_1(x_1) + f_2(x_2) + f_{12}(x_1, x_2). \tag{9}$$

Let $i \in \{1, 2\}$, define $\delta_i(\mathbf{x}^1, \mathbf{x}^2) := f_i(x_i^2) - f_i(x_i^1)$ and $\delta_{12}(\mathbf{x}^1, \mathbf{x}^2) := f_{12}(x_1^2, x_2^2) - f_{12}(x_1^1, x_2^1)$, the Shapley values $\psi$ for the difference $\delta$ is: $\psi_i(\mathbf{x}^1, \mathbf{x}^2) = \delta_i(\mathbf{x}^1, \mathbf{x}^2) + \frac{1}{2}\delta_{12}(\mathbf{x}^1, \mathbf{x}^2)$. Note that since $f$ is a Gaussian process, the Shapley value defined above is a random variable. Denote the posterior $f_u \sim \mathcal{GP}(\mu_u, \tilde{k}_u)$ for each $u \subseteq \{1, 2\}$, then the mean of the Shapley value is

$$\mathbb{E}[\psi_i] = \mu_i(x_i^2) - \mu_i(x_i^1) + \frac{1}{2}\left(\mu_{12}(x_1^2, x_2^2) - \mu_{12}(x_1^1, x_2^1)\right). \tag{10}$$

One can then sum over the data points to get aggregated (coarser) level attribution. The reasoning can be extended to the general cases with $d > 2$, see Appendix B for details.

Take the toy example in Section 4.1 for example. Suppose the input data $x_1$ now follows a different distribution $x_1 \sim \mathcal{N}(1, 0.25)$ that results in different outputs as shown in the left two plots in Figure 4 (blue versus yellow), the Shapley value attributing the delta between two output predictions can be found in Figure 4 (right) where we observe the output change is almost all due to the first feature $x_1$, aligning with our expectation. Not only is the mean of Shapley value attribution analytic, the variance also has a closed form following uncertainty propogation of Gaussian process models, see details in Appendix C and D. The confidence interval of attribution with $\pm 1$ standard deviation is represented with the black horizontal bar in Figure 4 (right), which captures the uncertainties from the OAK model and the noise in the data generating process.

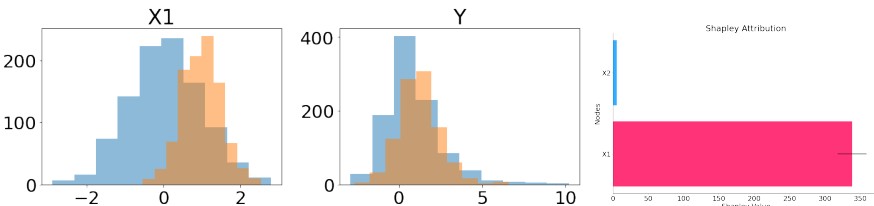

Figure 4: Data distributions for $x_1$ (left) and output $y$ (middle) for two datasets, represented in blue and orange respectively. Shapley value attributing the changes of outputs to changes in inputs (right).

## 5. Conclusion

In this paper, we tackle multiple challenging attribution problems in industrial systems through an explainable Gaussian process model. In particular, we extend the OAK model to applications in various attribution tasks, including attributing output/output changes to inputs/input changes and system change. We illustrate with toy examples on the benefits of applying the OAK model to attribution tasks due to its explainability, efficient computation and uncertainty quantification capability. Future work includes better experimental design for downstream decision making processes, such as automatically selecting features, guiding model development and active learning in picking most informative training data given attribution results.

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

## Appendix A. Orthogonal Binary Kernel

We define the kernel of a binary feature $x$ as

$$k(x, x') = \begin{cases} a & x = x' = 0 \\ b & x = x' = 1 \\ c & x \neq x' \end{cases} \tag{11}$$

A Gaussian Process can be defined as $f \sim \mathcal{GP}(0, k)$. Suppose the data distribution $p(x = 0) = p_0, \ p(x = 1) = p_1$, we say the kernel is orthogonal if

$$A := \sum_{x=0, x=1} f(x)p(x) = 0. \tag{12}$$

**Claim**: let

$$k(x, x') = \sigma^2 \begin{cases} p_1^2 & x = x' = 0 \\ p_0^2 & x = x' = 1 \\ -p_0 p_1 & x \neq x' \end{cases} \tag{13}$$

where $\sigma^2$ is the variance parameter, then $f \sim \mathcal{GP}(0, k)$ is orthogonal. To see this, we need to show that $A := \sum_{x=0, x=1} f(x)p(x) = 0$, this is equivalent to show

$$\mathbb{E}[A] = Var[A] = 0 \tag{14}$$

**Proof**

$$\mathbb{E}[A] = \sum_{x=0, x=1} \mathbb{E}[f(x)]p(x) = 0. \tag{15}$$

$$\begin{aligned} Var[A] = \mathbb{E}[A^2] &= \sum_{x=0, x=1} \sum_{y=0, y=1} \mathbb{E}[f(x)f(y)]p(x)p(y) \\ &= \sum_{x=0, x=1} \sum_{y=0, y=1} k(x, y)p(x)p(y) \\ &= \sigma^2(p_1^2 p_0^2 + p_1^2 p_0^2 - 2p_1^2 p_0^2) = 0. \end{aligned} \tag{16}$$

∎

## Appendix B. Shapley Value with $d > 2$.

Assuming the highest order of interaction to be 2. Then the Shapley value for feature 1 is

$$\phi_1 = \delta_1 + \frac{1}{2}\delta_{12} + \frac{1}{2}\delta_{13} + \cdots + \frac{1}{2}\delta_{1d}. \tag{17}$$

Its expectation is therefore

$$\mathbb{E}[\phi_1] = \mu_1(x_1^2) - \mu_1(x_1^1) + \frac{1}{2}\left(\mu_{12}(x_1^2, x_2^2) - \mu_{12}(x_1^1, x_2^1)\right) + \frac{1}{2}\left(\mu_{13}(x_1^2, x_3^2) - \mu_{13}(x_1^1, x_3^1)\right)$$
$$+ \frac{1}{2}\left(\mu_{1d}(x_1^2, x_d^2) - \mu_{1d}(x_1^1, x_d^1)\right). \tag{18}$$

The formula for other features have a similar form.

## Appendix C. Uncertainties of Attribution

We can quantify uncertainties of the above Shapley value analytically using the covariance of the posterior GP. Take $\phi_1$ for example, recall that $\mathrm{var}[X+Y] = \mathrm{var}[X] + \mathrm{var}[Y] + 2\mathrm{cov}[X,Y]$, we have

$$\mathrm{var}_{f_1}[\phi_1] = \mathrm{var}_{f_1}[\delta_1] + \frac{1}{4}\mathrm{var}_{f_{12}}[\delta_{12}] + \mathrm{cov}_{f_1, f_{12}}[\delta_1, \delta_{12}]. \tag{19}$$

For each $u \subseteq \{1, 2\}$,

$$\mathrm{var}_{f_u}[\delta_u] = \mathbb{E}_{f_u}[(f_u(\mathbf{x}^2) - f_u(\mathbf{x}^1))^2] - \left(\mathbb{E}_{f_u}[\mathbf{x}^2] - \mathbb{E}_{f_u}[\mathbf{x}^1]\right)^2 \tag{20}$$
$$= \mathbb{E}_{f_u}[(f_u(\mathbf{x}^2)^2 + f_u(\mathbf{x}^1)^2 - 2f_u(\mathbf{x}^1)f_u(\mathbf{x}^2))] - \left(\mu_u(\mathbf{x}^2) - \mu_u(\mathbf{x}^1)\right)^2 \tag{21}$$
$$= \tilde{k}_u(\mathbf{x}^1, \mathbf{x}^1) + \tilde{k}_u(\mathbf{x}^2, \mathbf{x}^2) - 2\tilde{k}_u(\mathbf{x}^1, \mathbf{x}^2). \tag{22}$$

For each $u, v \subseteq \{1, 2\}$,

$$\mathrm{cov}_{f_u, f_v}[d_u, d_v] = \mathrm{cov}_{f_u, f_v}[f_u(\mathbf{x}^2) - f_u(\mathbf{x}^1), f_v(\mathbf{x}^2) - f_v(\mathbf{x}^1)] \tag{23}$$
$$= \mathrm{cov}_{f_u, f_v}[f_u(\mathbf{x}^2), f_v(\mathbf{x}^2)] - \mathrm{cov}_{f_u, f_v}[f_u(\mathbf{x}^2), f_v(\mathbf{x}^1)] \tag{24}$$
$$- \mathrm{cov}_{f_u, f_v}[f_u(\mathbf{x}^1), f_v(\mathbf{x}^2)] + \mathrm{cov}_{f_u, f_v}[f_u(\mathbf{x}^1), f_v(\mathbf{x}^1)]. \tag{25}$$

For each of the covariance term, note that $f_u$ and $f_v$ are independent GP a priori, but the posterior GPs are no longer independent. Denote the prior GP of $f_u$ and $f_v$ to be $f_u^0 \sim \mathcal{GP}(0, k_u)$ and $f_v^0 \sim \mathcal{GP}(0, k_v)$ respectively, the posterior covariance can be calculated as

$$\mathrm{cov}_{f_u, f_v}[f_u(\mathbf{x}^a), f_v(\mathbf{x}^b)] = \mathrm{cov}_{f_u^0, f_v^0}[f_u^0(\mathbf{x}^a), f_v^0(\mathbf{x}^b)] \tag{26}$$
$$- \mathrm{cov}_{f_u^0}[f_v^0(\mathbf{x}^a), f_u^0(X)](K + \sigma^2 I)^{-1}\mathrm{cov}_{f_v^0}[f_v^0(X), f_v^0(\mathbf{x}^b)] \tag{27}$$
$$= \begin{cases} -k_u(\mathbf{x}^a, X)(K + \sigma^2 I)^{-1}k_v(X, \mathbf{x}^b) & u \neq v \\ k_u(\mathbf{x}^a, \mathbf{x}^b) - k_u(\mathbf{x}^a, X)(K + \sigma^2 I)^{-1}k_v(X, \mathbf{x}^b) & u = v \end{cases} \tag{28}$$

for any $a, b \in \{1, 2\}$, where $X$ is the training data and $K = \prod_{u \subseteq \{1,2\}} k_u(X_u, X_u)$ is the training input covariance across all inputs. The analysis can be generalised for dimension

$d > 2$, see details in Appendix C. The derivation can be extended to the aggregated level: suppose one is interested in attributing the sum of $N$ data points, we redefine $\delta_u$ as the sum of the delta across the data points. For each $u, v \subseteq \{1, 2\}$,

$$\text{cov}_{f_u, f_v}[\delta_u, \delta_v] = \text{cov}_{f_u, f_v}[\sum_{n=1}^{N} f_u(\mathbf{x}_n^2) - \sum_{n=1}^{N} f_u(\mathbf{x}_n^1), \sum_{n=1}^{N} f_v(\mathbf{x}_n^2) - \sum_{n=1}^{N} f_v(\mathbf{x}_n^1)] \quad (29)$$

where

$$\text{cov}_{f_u, f_v}[\sum_{n=1}^{N} f_u(\mathbf{x}_n^a), \sum_{n=1}^{N} f_v(\mathbf{x}_n^b)] = \sum_{n=1}^{N} \sum_{n'=1}^{N} \text{cov}_{f_u, f_v}[f_u(\mathbf{x}_n^a), f_v(\mathbf{x}_{n'}^b)] \quad (30)$$

$$= \begin{cases} -\sum_{n,n'=1}^{N} k_u(\mathbf{x}_n^a, X)(K + \sigma^2 I)^{-1} k_v(X, \mathbf{x}_{n'}^b) & u \neq v \\ \sum_{n,n'=1}^{N} k_u(\mathbf{x}_n^a, \mathbf{x}_{n'}^b) - \sum_{n,n'=1}^{N} k_u(\mathbf{x}_n^a, X)(K + \sigma^2 I)^{-1} k_v(X, \mathbf{x}_{n'}^b) & u = v \end{cases} \quad (31)$$

## Appendix D. Uncertainties of Attribution for $d > 2$

$$\text{var}(\phi_i) = \frac{1}{n^2} \sum_{S \subseteq \{x_1, \cdots x_n\} \setminus \{x_i\}} \sum_{T \subseteq \{x_1, \cdots x_n\} \setminus \{x_i\}} \binom{n-1}{|S|}^{-1} \binom{n-1}{|T|}^{-1}$$
$$\times \text{cov}[v(S \cup \{i\}) - v(S), v(T \cup \{i\}) - v(T)] \quad (32)$$

where

$$\text{cov}[v(S \cup \{i\}) - v(S), v(T \cup \{i\}) - v(T)] = \text{cov}[v(S \cup \{i\})v(T \cup \{i\})]$$
$$- \text{cov}[v(S \cup \{i\}), v(T)]$$
$$- \text{cov}[v(S), v(T \cup \{i\})] + \text{cov}[v(S), v(T)] \quad (33)$$

and each of the term can be calculated as

$$\text{cov}[v(S), v(T)] = \text{cov}[\sum_{u \subseteq S} \delta_u, \sum_{v \subseteq T} \delta_v] = \sum_{u \subseteq S} \sum_{v \subseteq T} \text{cov}[\delta_u, \delta_v] \quad (34)$$

where each of the term in the summation can be computed by (25).

