# OpenReview forum: "Explainable Attribution using Additive Gaussian Processes"
_approximateinference.org/AABI/2024/Symposium — AABI 2024_

### Official Review · Reviewer_NPT5 · 2024-04-08
**The paper introduces a novel methodology leveraging additive Gaussian processes for attributing outputs to inputs in industrial systems, and addressing various attribution tasks with theoretical grounding and practical demonstrations.**

**Rating:** 6
**Confidence:** 4

**Review:**

Strengths:
1. The paper rigorously establishes the theoretical foundation of explainable attribution using additive Gaussian processes (AGPs). Leveraging AGPs allows for decomposing complex system outputs into interpretable components, enhancing the reliability and accuracy of attribution analyses. The use of clear language, structured presentation, and illustrative examples facilitates understanding, enabling readers to grasp the nuances of AGP-based attribution.
2. The empirical demonstrations in the paper effectively illustrate the practical relevance and efficacy of the proposed AGP-based attribution framework in industrial systems. By applying the methodology to real-world scenarios, the paper demonstrates its utility in providing transparent and interpretable explanations for system updates, input changes, and output evolution.

Weaknesses:
1. The paper lacks a comprehensive discussion on potential limitations or scalability concerns associated with the AGP-based attribution methodology. Addressing issues such as computational complexity, scalability to larger datasets, or applicability to diverse industrial contexts would provide readers with a more nuanced understanding of the practical constraints and challenges.
2. The scalability of the approach to larger datasets or more complex industrial systems remains uncertain. Further exploration and validation across diverse applications are needed to assess the methodology's scalability, generalizability, and robustness in real-world settings.
3. The empirical validation primarily relies on toy examples and synthetic data, which may not fully capture the complexities and nuances of real-world industrial systems. Without extensive external validation on diverse and representative datasets from actual industrial applications, the generalizability and robustness of the proposed AGP-based attribution framework remain uncertain.

---

### Official Review · Reviewer_AQ6m · 2024-04-13
**This paper presents a novel explainable attribution framework based on additive Gaussian processes, with strong technical merits and practical relevance.**

**Rating:** 8
**Confidence:** 4

**Review:**

Pros:
1. Originality and Significance: The paper tackles important and challenging problems in industrial settings where complex black-box systems need to be interpreted. The proposed approach based on additive Gaussian processes with orthogonal additive kernels (OAK) is a novel contribution that combines explainability, accurate modeling, and efficient computation of attribution metrics like Shapley values.

2. Methodological Rigor: The paper provides a solid theoretical foundation, deriving closed-form expressions for Sobol indices and Shapley values under the OAK framework. The handling of system changes and output changes is also well-designed, with a principled way of incorporating additional binary features and leveraging the OAK structure.

3. Practical Relevance: The authors illustrate the applicability of their method using synthetic examples for each of the three attribution tasks. The results demonstrate the effectiveness of the OAK model in recovering the true underlying functional decomposition and providing meaningful and interpretable attribution results.

4. Clarity: The paper is well-structured and the explanations are clear. The authors provide sufficient details on the OAK model and the attribution calculations, making the work accessible to readers familiar with Gaussian processes and Shapley values.

Cons:
1. Lack of Real-world Evaluation: While the synthetic examples are helpful for illustrating the proposed approach, the paper would benefit from evaluating the method on real-world industrial datasets to demonstrate its practical impact and effectiveness in tackling complex, high-dimensional problems.

2. Limited Scope: The paper focuses on three specific attribution tasks, but there might be other types of attribution problems encountered in practice that are not covered. Expanding the scope or discussing the generalizability of the approach would strengthen the paper.

3. Computational Complexity: The analytical expressions for Sobol indices and Shapley values are a strength of the OAK model, but the computational complexity might still be a limitation for very high-dimensional problems. The authors could discuss the scalability of their method and potential strategies to address this challenge.

---

### Official Review · Reviewer_LgAu · 2024-04-21
**Additional work seems to be needed**

**Rating:** 3
**Confidence:** 3

**Review:**

**- Summary**
This paper proposes an explainable attribution framework based on additive Gaussian processes with an Orthogonal Additive Kernel.

**- Pros**
By recognizing the relationship between Sobol’ Index and Shapley Value (Owen 2014), the authors adapt the OAK framework (Lu et al., 2022) to obtain Shapley values, thereby explaining feature attributes of a black-box system.

**- Cons**
Although the use of additive Gaussian processes with an Orthogonal Additive Kernel (OAK, Lu et al., 2022) appears novel, the paper presents limited new results, and further work is expected.

1. Known results:
  - Section 2 is mainly from Lu et al., 2022;
  - Section 3 is mainly from Owen 2014;
  - Section 4.1 appears to run the OAK model on a toy example.

2. New results but more work is expected:
- Section 4.2 introduces the Orthogonal Binary Kernel, which indeed appears to be a novel adaptation of the OAK model. However, the original OAK model should also work as this scenario could be included as a part of the OAK framework with $x_3 = \omega$. A comparison between using the original OAK with $x_3 = \omega$ and the proposed version using Orthogonal Binary Kernel would be insightful. Additionally, an application to real data would effectively demonstrate the utility of this scenario—although the authors describe potential practical applications, the motivation for this type of model change is not very straightforward.

- Section 4.3 proposes to use Equation (8), but it is difficult to see the advantage of using this equation for explaining the contribution of features. It would be beneficial if the authors could add a performance comparison between the proposed metric and other methods, using either synthetic examples or real data. Furthermore, the rationale for introducing Equation (8) seems abrupt. Are there any motivations or references for using this metric?


3. Computational analysis
As mentioned in Section 1, computational efficiency often plays a critical role in practice. Could the authors add a time cost evaluation and compare the proposed methods with other Shapley value methods [for both large samples (n) and, more importantly, large numbers of features (d)]?

**- Conclusion**
In conclusion, I think this is an interesting idea going from OAK to Shapley Value, however more work is needed as mentioned above, especially in the computational analysis with respect to the number of features. Equations (1) and (2) seem to suggest that scalability with respect to d could lead to some concerns in practice.

---

### Official Review · Reviewer_iVDx · 2024-04-24
**Extending OAK for specific attribution cases**

**Rating:** 5
**Confidence:** 4

**Review:**

This work leverages additive Gaussian processes as proposed by Lu et al. (2022) to propose an explainable attribution framework, by focusing on three major types of attribution tasks commonly seen in practice.

The authors extend the OAK model to include attributing output changes to input changes and to systems change. However, it is unclear how significant the contributions of this work are, due to
1. the direct use of the methodology presented in Lu et al. (2022), and
2. the demonstration being limited to toy examples for each use case.

More precisely, the authors state that their contributions include:

1. to "propose to use Sobol indices (Owen, 2014), and we provide closed-form expressions for the Sobol indices and Shapley value under the OAK Gaussian process model" and  "provide analytic expressions for Shapley values under the OAK model"
- However, the original work of Lu et Al. (2022) already states that "the Sobol indices are analytic for the OAK model" and present them in their work, as acknowledged by the authors in Section 3. Hence, one wonders what exactly is the contribution of the authors here

2. to "propose to model the dataset shift using an additional indicator input variable. We provide a kernel under the OAK framework for handling this additional (binary) input variable in a way that is consistent with continuous OAK kernels.".
- This orthogonal binary kernel is helpful for their work in Section 4.2 in attributing Output to Inputs with System Change, which is limited to binary systems: old ($w=0$) or new ($w=1$).

Presented results are limited to several toy examples, showcasing limited value of the proposed use of OAK kernels for attribution.

---

### Official Review · Reviewer_c2ew · 2024-04-24
**Sobol' indices and Shapley value connected via ANOVA to attribute outputs to inputs in orthogonal additive kernel models**

**Rating:** 7
**Confidence:** 3

**Review:**

This submission builds upon the recently developed orthogonal additive kernel (OAK) model (Lu 2022 PMLR) to develop a framework for attributing outputs to inputs based on the Shapley value. The key to enabling this is theorem one from Owen (Journal of Uncertainty Quantification 2014) connecting Sobol' index and Shapley value via an ANOVA decomposition which has a simple form in the OAK model. (The connection is slightly obscured by the fact that Owen discusses two Sobol’ indices, only one of which is the above referenced Sobol’ index).

Because Shapley value can be easily calculated in the OAK model this is leveraged to consider three scenarios 1) attributing inputs to outputs, 2) attributing inputs to outputs with system change, and 3) attributing output changes to inputs changes. For 1) it's a straightforward application of the theory and the results for the most part are as I would expect. There is a small discrepancy in the left panel of figure 1 with the explanation "Note that the constant gap between the truth and OAK in the first plot is expected and is captured with the constant kernel." While space is limited a little more exposition of this phenomena would be appreciated. For 2) the model of 1) is extended with a binary indicator variable w indicating w=0 the "old system" and w=1 the "new system". The results of figure 3 show the modified model can be handled as well, albeit with an (expected) increase in the variance. For 3) the model of 1) is extended to account for variations in the input distributions, leading to an extension of the Sobol’ index as value function. It is then demonstrated that a perturbuation of the input distribution for $x_1$ can be detected.

Generally the submission is clearly written. There are a few places with awkward language, but these don't seem particularly important for a potential poster based on this abstract. The name "Sobol’" is written as both "Sobol" and "Sobol’", with my understanding that the proper name includes the apostrophe. The references are at times incomplete, e.g. the Shapley reference is missing any journal/volume/series information. These are also minor issues for a submission not meant for publication per se, but could be corrected for a poster.

Overall this is an interesting submission. In some respects the results are straightforward extensions of the Lu 2022 results on the OAK model, but they do point the way towards potentially interesting useful extensions of that work not previously explored. While not explicitly Bayesian in nature, it may be of interest to AABI attendees and could be seen in some ways as a Bayesian approach to causal inference.

---

### Meta-Review · Area_Chair_ArQ6 · 2024-05-27

**Recommendation:** Accept (Poster)
**Confidence:** 4

**Metareview:**

The authors present an approach for explainable attribution in situations where black boxes need to be interpreted. The approach combines additive GPs with orthogonal additive kernels for accurate modeling that is also computationally efficient. While the reviewers agree the submission was generally well written, there were a number of concerns about the results and the fact that experimentation was limited in terms of real world evaluation/not capturing the nuances of real world systems. Nonetheless, the paper does establish some theoretical foundations for explainable attribution so I vote for accepting this paper.

---

### Decision · Program_Chairs · 2024-05-27

Accept